# MicroRNA-30d-5p—A Potential New Therapeutic Target for Prevention of Ischemic Cardiomyopathy after Myocardial Infarction

**DOI:** 10.3390/cells12192369

**Published:** 2023-09-27

**Authors:** Elke Boxhammer, Vera Paar, Bernhard Wernly, Attila Kiss, Moritz Mirna, Achim Aigner, Eylem Acar, Simon Watzinger, Bruno K. Podesser, Roland Zauner, Verena Wally, Michael Ablinger, Matthias Hackl, Uta C. Hoppe, Michael Lichtenauer

**Affiliations:** 1Internal Medicine II, Department of Cardiology, Paracelsus Medical University Salzburg, 5020 Salzburg, Austria; e.boxhammer@salk.at (E.B.);; 2Department of Internal Medicine, General Hospital Oberndorf, Teaching Hospital of the Paracelsus Medical University, 5110 Oberndorf, Austria; 3Ludwig Boltzmann Cluster for Cardiovascular Research, Center for Biomedical Research and Translational Surgery, Medical University Vienna, 1090 Vienna, Austria; attila.kiss@meduniwien.ac.at (A.K.);; 4Rudolf Boehm-Institute for Pharmacology and Toxicology, Clinical Pharmacology, Leipzig University, 04107 Leipzig, Germany; achim.aigner@medizin.uni-leipzig.de; 5Dermatology, Paracelsus Medical University Salzburg, 5020 Salzburg, Austria; 6TAmiRNA GmbH, Muthgasse 18, 1110 Vienna, Austria

**Keywords:** cardioprotection, ischemic cardiomyopathy, miR-30d-5p, myocardial infarction

## Abstract

(1) Background and Objective: MicroRNAs (miRs) are biomarkers for assessing the extent of cardiac remodeling after myocardial infarction (MI) and important predictors of clinical outcome in heart failure. Overexpression of miR-30d-5p appears to have a cardioprotective effect. The aim of the present study was to demonstrate whether miR-30d-5p could be used as a potential therapeutic target to improve post-MI adverse remodeling. (2) Methods and Results: MiR profiling was performed by next-generation sequencing to assess different expression patterns in ischemic vs. healthy myocardium in a rat model of MI. MiR-30d-5p was significantly downregulated (*p* < 0.001) in ischemic myocardium and was selected as a promising target. A mimic of miR-30d-5p was administered in the treatment group, whereas the control group received non-functional, scrambled siRNA. To measure the effect of miR-30d-5p on infarct area size of the left ventricle, the rats were randomized and treated with miR-30d-5p or scrambled siRNA. Histological planimetry was performed 72 h and 6 weeks after induction of MI. Infarct area was significantly reduced at 72 h and at 6 weeks by using miR-30d-5p (72 h: 22.89 ± 7.66% vs. 35.96 ± 9.27%, p = 0.0136; 6 weeks: 6.93 ± 4.58% vs. 12.48 ± 7.09%, *p* = 0.0172). To gain insight into infarct healing, scratch assays were used to obtain information on cell migration in human umbilical vein endothelial cells (HUVECs). Gap closure was significantly faster in the mimic-treated cells 20 h post-scratching (12.4% more than the scrambled control after 20 h; *p* = 0.013). To analyze the anti-apoptotic quality of miR-30d-5p, the ratio between phosphorylated p53 and total p53 was evaluated in human cardiomyocytes using ELISA. Under the influence of the miR-30d-5p mimic, cardiomyocytes demonstrated a decreased pp53/total p53 ratio (0.66 ± 0.08 vs. 0.81 ± 0.17), showing a distinct tendency (*p* = 0.055) to decrease the apoptosis rate compared to the control group. (3) Conclusion: Using a mimic of miR-30d-5p underlines the cardioprotective effect of miR-30d-5p in MI and could reduce the risk for development of ischemic cardiomyopathy.

## 1. Introduction

Ischemic cardiomyopathy (CMP) after acute myocardial infarction (MI) is the most common cause of heart failure and therefore the leading cause of death worldwide [1,2]. While in the acute phase of MI, percutaneous transluminal angioplasty (PTCA) is the most important means to restore coronary perfusion (“time is muscle”) [3], lifestyle modification as well as the drug regimes of anti-platelet therapy, cholesterol lowering and heart failure therapy [4,5] play a crucial role in recurrence prevention, maintenance of cardiac functional performance as well as prevention of adverse ventricular remodeling.

At the present time, the risk with respect to the development of ischemic CMP depends, among other factors, on the infarct location and on the duration of complete reperfusion of the affected coronary vessel [6,7,8,9]. The irreversible destruction of cardiomyocytes leads to the development of a corresponding infarct area with loss of myocardial contractility due to scarring [10,11]. All current drug therapy options aim at preserving the remaining myocardial functional capacity. Therapeutic options to prevent or reduce cell death of cardiomyocytes or even to regenerate them after cell death are still lacking.

The broad research field of microRNAs (miRs) could offer potential new therapeutic approaches to prevent ischemic CMP by acting on the post-transcriptional regulation of corresponding gene expressions [12,13,14]. Despite their small size of approximately 20 nucleotides, miRs, as small non-coding RNAs, have a significant impact on cell proliferation and differentiation as well as apoptosis regulation. In particular, miRs play a relevant role in the development of blood vessels, the contractility and regeneration of cardiomyocytes, the development and stabilization of arteriosclerosis, and the regulation of the basic cardiac rhythm in the context of various underlying cardiovascular diseases [15,16,17].

In the setting of acute MI, miRs may not only be considered biomarkers for improved early diagnosis [18,19,20], but also act as clinical prognostic markers for the occurrence of major cardiovascular events (MACEs) or mortality [21]. However, miRs can not only be quantified but also modified to increase (agomir/mimic) or decrease (antagomir) expression [22].

Based on these considerations, miRs have also been used therapeutically in previous work in both in vivo and in vitro studies. For example, myocardial ischemia was induced using animal models and then agomirs/mimics of miR-21 [23,24], miR-199a or miR-590 [25,26] were administered, which stimulated cardiac repair. The use of antagomirs of miR-1 [27], miR-24 [28] and miR-132 [29,30] in comparable animal models also improved cardiac functional performance after induced ischemia.

The aim of this experimental work was to establish an miR expression profile of ischemic vs. healthy myocardium using an animal model as a first step. Based on these results, miR-30d-5p, which was significantly downregulated in the setting of acute MI, was considered as a potential therapeutic target. In a second step, in vivo and in vitro studies were used to investigate the effect of “drug” stimulation of miR-30d-5p on the development of ischemic CMP.

## 2. Materials and Methods

### 2.1. Experimental Rat Model

#### 2.1.1. Induction of Acute Myocardial Infarction

The animal experiment described in this publication was approved by the Committee for Animal Research of the Medical University of Vienna 66.009/0122-WF/V/3b/2017 and was performed in accordance with the Guide for the Care and Use of Laboratory Animals by the National Institutes of Health (NIH Publication No. 85-23, revised 1996).

Male Sprague-Dawley rats were used for the animal model and were anesthetized by intraperitoneal injection of xylazine (4 mg/kg; Bayer, Leverkusen, Germany) and ketamine (100 mg/kg; Dr. E. Gräub AG, Bern, Switzerland). Ventilation with a 14-gauge tube was performed at 9 mL/kg body weight and a respiratory rate of 75–85 stroke/min. Adequate analgesia was provided by intraperitoneally injected piritramide (0.1 mL/kg body weight). Rectal temperature was monitored and maintained at 37.5–38.5 °C by a heated operating table, and heart rhythm was monitored by electrocardiogram (ECG). Myocardial infarction was induced by coronary artery ligation as described previously [31,32]. To induce myocardial ischemia, the heart was exposed via a left thoracotomy and a permanent ligation of the left coronary artery (LCA) 2–3 mm away from the origin using a 6–0 prolene. Myocardial ischemia was documented by ECG through the development of ST segment elevations. For postoperative analgesia, piritramide was administered via drinking water (2 ampules of Piritramide with 30 mL of Glucose 5% in 250 mL water). In order to obtain the respective histological samples of the hearts, the rats were sacrificed after six weeks under deep anesthesia by opening the inferior vena cava.

#### 2.1.2. MiR Expression with Next-Generation Sequencing (NGS)

MiR profiling was performed by NGS to assess differences in miR expression in ischemic vs. healthy myocardium in the animal model mentioned above (Figure 1A). A total of 13 animals were subjected to the surgical intervention. Eight of them underwent ischemia injury via ligation of the LCA, while five sham-operated animals without ligation of the LCA served as a control group. Six weeks after the induction of MI, the rats were sacrificed and the hearts were explanted and immediately deep frozen in liquid nitrogen and kept at −80 °C.

##### Total RNA Extraction from Tissue

Total RNA from frozen tissues was extracted using an miRNeasy Mini Kit (Qiagen, Hilden, Germany). In the first step, fresh frozen tissue was weighed and subsequently 350 µL Qiazol was added. The subsequent steps were performed at room temperature. TissueRuptor homogenizer (Qiagen, Hilden, Germany) was applied to each tissue for 30 s at full speed until no visible tissue parts were observed. After an incubation at room temperature for 5 min, 140 µL chloroform was added to the lysates followed by vigorous mixing in a vortex, incubation at room temperature for 3 min and cooled centrifugation at 12,000× *g* for 15 min at 4 °C. Exactly 350 µL of the upper aqueous phase was mixed with 525 µL ethanol and RNA was precipitated on an miRNeasy mini column followed by automated washing with RPE and RWT buffer in a Qiacube liquid handling robot (Qiagen, Hilden, Germany). Finally, total RNA was eluted in 30 µL nuclease-free water. Total RNA integrity and concentration were checked using the RNA6000 Nano Bioanalyzer assay (Agilent, Santa Clara, CA, USA) as well as spectrophotometric RNA quantification (Nanodrop; Thermo Fisher Scientific, Waltham, MA, USA).

##### Small RNA Sequencing

Equal amounts of total RNA (230 ng) were used for small RNA library preparation using an NEBNext small RNA library preparation kit (New England Biolabs, Ipswich, MA, USA) according to the manufacturer’s instructions. Adapter-ligated libraries were amplified using barcoded Illumina reverse primers in combination with the Illumina forward primer. Libraries were pooled equimolar on the basis of DNA 1000 Bioanalyzer quantification (Agilent, Santa Clara, CA, USA) and gel-based size selection was performed on the pooled library to enrich for insert sizes between 18 and 36 base pairs. The library pool(s) were quantified using the qPCR KAPA Library Quantification Kit (KAPA Biosystems; Roche Holding, Basel, Switzerland). The library pool(s) were then sequenced on a NextSeq500 sequencing instrument (50 bp single end) according to the manufacturer’s instructions, yielding an average of 11.9 million reads per sample (10.6M to 12.8M). Raw data were de-multiplexed and FASTQ files for each sample were generated using the bcl2fastq software, v1.8.4 (Illumina Inc., San Diego, CA, USA). FASTQ data were checked using the FastQC tool (http://www.bioinformatics.babraham.ac.uk/projects/fastqc/, accessed on 16 October 2019).

##### Small RNA Sequencing Data Analysis

Cutadapt (1.11) was used to extract, trim and remove adapters. Bowtie2 (2.2.2) was used for mapping the reads, requiring a perfect match to the reference miR base sequences and allowing for one mismatch in the first 32 bases of the read for the respective genome mappings.

#### 2.1.3. Histology and Planimetry

Once miR-30d-5p was established as a promising target and thus a potential cardioprotective agent, the infarct rat model was used.

To assess the effect of miR-30d-5p on the acute ischemic response, 17 rats were subjected to myocardial infarction. Ten rats received a miR-30d-5p mimic (Dharmacon, Lafayette, LA, USA) intraperitoneally, whereas seven rats were administered a non-functional scrambled siR (Dharmacon, Lafayette, LA, USA), also intraperitoneally (Figure 2A,B). The rats were sacrificed 72 h after the coronary artery ligation.

In addition, 19 animals (9 with application of the miR-30d-5p mimic and 10 with application of the scrambled siR) were used to evaluate the myocardial infarct area 6 weeks after MI induction (Figure 3A,B). The necessary miR/siR was packaged into appropriate extracellular vesicles as described in the study of Borchardt et al. [33]. A quantity of 17.7 µg (6.65 µL of a 200 µM stock solution) miR mimic or scrambled siR was dissolved in 200 µL HN buffer (10 mM HEPES, 150 mM NaCl, pH 7.4) and incubated for 10 min [34,35]. In parallel, 133 µg PEI F25-LMW was diluted in 200 µL HN buffer and incubated for 10 min prior to mixing with the miR solution and complexation for 15 min. The nanoparticles were stored at −80 °C. For injection, aliquots were thawed and kept at room temperature for 30 min prior to use. After intraperitoneal injection, these exosome-like extracellular vesicles containing miR-30d-5p enter the bloodstream or interstitial fluid. Cardiomyocytes may take up these extracellular vesicles, allowing the enclosed miR-30d-5p to exert its regulatory effects within these cells [36].

After 6 weeks, the hearts were explanted and the organs were sliced at three layers at the level of the largest extension of infarcted area. Sections were fixed in 10% neutral buffered paraformaldehyde, embedded in paraffin and sliced into 5 µm sections. Subsequently, both hematoxylin–eosin (HE) staining and van Gieson (VG) staining were performed. An inverted microscope with 200× and 400× magnification was applied to evaluate the tissue specimen. Image J planimetry software, version 1.53k (Rasband, W.S., Image J, U.S. National Institutes of Health, Bethesda, MD, USA) was utilized to assess the extent of the necrosis/fibrotic area after 6 weeks. Infarction size was expressed as a percentage of the total left ventricular area as described previously [37].

### 2.2. Human Umbilical Cord Endothelial Cells (HUVECs)

#### 2.2.1. Cultivation and Transfection with Nanoparticles of the miR Mimic or Scrambled siR

HUVECs (Thermo Fisher Scientific, Waltham, MA, USA) were cultivated in T75 flasks using endothelial cell growth medium 200 (M200500, ThermoFisher Scientific, Waltham, MA, USA). Afterwards, 1.5 × 10^5^ cells in 500 µL cultivation medium were seeded in Costar^®^ 24-well plates (CLS3527, Corning Costar, ThermoFisher Scientific, Waltham, MA, USA) and incubated for 24 h at 37 °C. The corresponding HUVECs were subjected to a transfection process when a confluence of nearly 80% was reached. The HUVEC transfection protocol on a 24-well plate was carried out on the basis of the manufacturer’s instructions by means of the XfectTM RNA Transfection Reagent Protocol. Therefore, an Xfect polymere (631450, Takara Bio Inc., Kusatsu, Japan) was used mixed with either the aforementioned nanoparticles of miR-30d-5p mimic or a scrambled siR. Mimic miR and scrambled siR were brought to a concentration of 50 nM and applied to the respective wells. The total incubation time with the transfection medium was 4 h. This was then replaced with normal endothelial growth medium and incubated for a further 44 h at 37 °C with 5% CO_2_. The growth medium was changed every 24 h.

#### 2.2.2. Detection of Infarct Healing via Cell Migration Scratch Assay

Forty-eight hours after treatment with the transfection medium, a 100 µL pipette tip was used to manually obtain a cell-free gap for the cell migration scratch assay. The growth processes over the scratch were monitored every hour for 24 h by measuring cell confluence in a pre-defined, constant area including the gap using a Tecan Spark^®^ 10M multimode microplate reader (Tecan, Männedorf, Switzerland). The percentage of gap closure was assessed 20 hrs after scratch initiation (t = 0) using image analysis software (ImageJ, version 1.53c, NIH, USA) to determine the gap area normalized to the gap area at timepoint t = 0 (Figure 4A).

### 2.3. Human Cardiomyocytes (HCMs)

#### 2.3.1. Cultivation and Transfection with Nanoparticles of the miR Mimic or Scrambled siR

The cultivation of HCMs (PromoCell, Heidelberg, Germany) was based on a myocyte growth medium supplemented with supplement mix (PromoCell, Heidelberg, Germany), 1% L-glutamine and 1% PenStrep in T75 flasks. For the following enzyme-linked immunosorbent assay (ELISA), cells were seeded at a concentration of 1 × 10^4^ cells in a 96-well plate (Greiner Bio-One; Frickenhausen, Germany) with 150 µL cultivation medium. Confluence was achieved at a cell density of 4 × 10^4^ cells per well after 24 h. If confluence was not visually achieved, the culture medium was replaced with fresh 150 µL cultivation medium. When confluence was satisfactory, the basal medium was removed and starvation medium (supplemented only with 1% PenStrep and 1% L-glutamine) was applied for 3 h instead. HCMs were further treated for transfection with the same protocol as used in HUVEC cells (Figure 5A). Again, nanoparticles including the miR-30d-5p mimic and a scrambled siR were applied in a concentration of 50 nM for 15 min.

#### 2.3.2. Apoptosis Testing by p53 and pp53 ELISA

P53 plays a relevant key role in programmed cell death. Due to numerous stimuli in the context of acute myocardial infarction and associated ischemic injury, hypoxia, oxidative stress and damage to cardiomyocyte DNA occur. Among other things, this leads to increased activation of the enzyme Ataxia-telangiectasia mutated (ATM) kinase, a serine protein kinase that initiates phosphorylation of p53. This phosphorylation causes the conformational state of p53 to change such that the inactive form originally bound to its regulator and proto-oncogen Mouse Double Minute 2 Homolog (MDM2) is released and thus activated. Through tetramerization, p53 reaches the configuration to finally initiate cell death at the nucleus level by means of increased transcription of corresponding genes. For better understanding, the molecular processes are shown in a schematic overview in Figure 5B.

To investigate the anti-apoptotic and thus simultaneously cardioprotective effect of miR-30d-5p, the abovementioned prepared HCMs were subjected to p53 (DuoSet^®^ IC ELISA—Human Total p53, R&D Systems, Abingdon, UK) and pp53 ELISA analysis (DuoSet^®^ IC ELISA—Human Phospho-p53, R&D Systems, Abingdon, UK). Accordingly, conclusions about the apoptosis rate of cells treated with and without the mimic of miR-30d-5p can be drawn from the ratio between phosphorylated and total p53. For this purpose, a 96-well plate was split, and one side was incubated with p53-capture antibody and the other with pp53-capture antibody according to the manufacturer’s instructions at room temperature overnight. Afterwards, the pretreated HCMs were detached from the 96-well plate with several washings and application of a lysis buffer containing 0.5% Triton X-100. The obtained solution was centrifuged and the cell supernatant was transferred to the prepared 96-well plate containing capture antibody. After the incubation time was observed, the supernatant was removed and either the biotinylated p53 or pp53 detection antibody was added. Finally, in two additional steps, streptavidin-Horseradish peroxidase and tetramethylbenzidine were added to achieve a color reaction. Subsequently, the analysis was performed using a microplate reader set to a wavelength of 450 nm.

### 2.4. Statistical Analysis

Statistical analysis of the NGS data was performed with the statistical software R (version 4.2.0). To identify microRNAs that were significantly regulated between ischemia and control animals, differential expression analysis was performed using the edgeR package v3.28 (Bioconductor, http://bioconductor.org/, accessed on 12 December 2019) and, in this case, specific log2-transformed fold change (log2(FC)). The log2(FC) describes the fold difference in microRNA levels between the control and treatment groups. Positive log2(FC) describes upregulation and negative log2(FC) describes downregulation in the treatment group.

The further statistical analysis was carried out using GraphPad Prism software (GraphPad Prism version 8.0.0, GraphPad Software, San Diego, CA, USA, www.graphpad.com, accessed on 16 December 2019) and SPSS 25 (IBM Corp. Released 2017. IBM SPSS Statistics, Version 25.0, Armonk, NY, USA), along with, again, the use of R (version 4.2.0). Normally distributed data are given as means ± standard deviations (SDs) and non-normally distributed data as medians ± interquartile ranges (IQRs). Categorical variables are expressed as percentages. In the case of a normal distribution and independent samples, an unpaired Student’s *t*-test was performed, and in the case of a non-normal distribution, a Wilcoxon–Mann–Whitney U test was conducted. In case of paired samples, a Wilcoxon signed-rank test was figured out. All tests were two-sided. A *p*-value ≤ 0.05 was considered statistically significant.

## 3. Results

### 3.1. miR-30d-5p with Significant Downregulation in the Animal Model of Ischemic CMP

In the context of NGS sequencing, 202 miRs were analyzed with respect to their expression between healthy myocardium and ischemic myocardium. An overview of all analyzed miRs is provided in Figure 1B. It can be seen that a large number of miRs were significantly regulated between the sham-operated and LCA-ligated rats. Therefore, a volcano plot was created for a better overview (Figure 1C). Here, it could be demonstrated that the majority of miRs in the ischemia group were significantly upregulated (log2(FC) values > 1), again with the top 20 upregulated miRs shown in Table 1. The highest log2(FC) values were achieved by miR-223-3p (3.78) and miR-212-3p (3.48), each with a highly significant *p*-value < 0.001. An additional, artificial increase in these already upregulated miRs after myocardial ischemia by drug delivery would therefore not lead to any additional benefit, so the focus of this study was rather on the downregulated miRs with a negative log2(FC) value < −0.5.

In this regard, Table 2 shows the top 20 downregulated miRs. Here, miR-328a-3p with a log2(FC) of −1.10 ranks as the most significantly downregulated miR of the entire cohort studied, followed by miR-125a-5p (log2(FC): −0.94). Like the other miRs, miR-30d-5p, which was further investigated in this work, provided highly significant downregulation with a log2(FC) value of −0.65.

In preliminary studies of our working group [38,39], we observed a complex regulation pattern of miR-30d-5p in response to different stimuli. Specifically, both in in vivo and in vitro studies, we found that stimulation with anti-thymocyte globulin (ATG) led to a significant upregulation of miR-30d-5p. This finding demonstrates the dynamic nature of miR-30d-5p expression in response to specific physiological and pathological conditions, including myocardial ischemia.

The decision to investigate miR-30d-5p further was not based solely on its downregulation in myocardial ischemia. Instead, it was driven by the dual nature of its regulation, showing potential for both downregulation in ischemic conditions and upregulation in response to ATG stimulation. This dual regulation pattern piqued our interest, as it suggests that miR-30d-5p may play a multifaceted role in the context of cardiac ischemia and may be responsive to therapeutic interventions.

Furthermore, our choice was informed by a comprehensive literature review conducted in prominent scientific databases, such as PubMed and Scopus. This review revealed that miR-30d-5p has been implicated in various cardiac processes, including pathways associated with myocardial infarction. While other miRNAs may indeed exhibit more significant downregulation in isolation, miR-30d-5p’s unique response to ATG stimulation, coupled with its established involvement in cardiac biology, made it a compelling candidate for our investigation.

### 3.2. Significant Reduction in Infarct areal Size in Animals Treated with the Mimic of miR-30d-5p after 72 h and after 6 Weeks

As described in the Materials and Methods section, an experimental rat model of myocardial infarction (ligation of LCA) was used to evaluate a potential therapeutic benefit of a mimic of miR-30d-5p regarding a potential reduction in myocardial scar size in comparison to the administration of a scrambled siR.

In the 72 h animals, those treated with miR-30d-5p showed a volumetric infarct size of 22.89 ± 7.66% on average, whereas those rats treated with scrambled siR presented an infarct extent of 35.96 ± 9.27%. Here, a statistically significant difference was found, with *p* = 0.0136.

Rat hearts 6 weeks after induced MI administered a mimic of miR-30d-5p showed a volumetric infarct area of 6.93 ± 4.58%, whereas scrambled siR-treated animals demonstrated an infarct size of 12.48 ± 7.09%. This revealed a statistically significant difference between the two groups, with a *p*-value of 0.0172.

Representative images can be seen in Figure 2C,D for the 72 h animals and in Figure 3C,D for the 6-week animals; Figure 2E and Figure 3E represent the respective boxplot diagrams as a summary.

### 3.3. Significantly Faster Gap Closure in Cell Migration Scratch Assay of HUVECs Transfected with the Mimic of miR-30d-5p

To investigative the healing process after myocardial infarction under the influence of a mimic of miR-30d-5p, the cell migration scratch assay mentioned above was performed. In total, three independent experiments with four wells per experimental group and run were conducted. A difference in % gap closure was evaluated with an unpaired, two-sided, non-parametric Wilcoxon signed-rank test on gap areas merged from all wells of all independent experiments. The results indicated that cells treated with the mimic for 48 h closed the gap significantly faster by 12.4% at 20 h post-scratching (*p*-value = 0.013). Representative images are shown in Figure 4B, while a boxplot diagram sums up the results of the unpaired Wilcoxon test (Figure 4C).

### 3.4. Reduced Apoptotic Rate in HCMs Transfected with the Mimic of miR-30d-5p

To understand the cardioprotective mechanism of miR-30d-5p also at the cellular level, after appropriate cultivation of HCMs and transfection with, again, a mimic of miR as well as a scrambled siR, a p53/pp53 ELISA was performed to determine the apoptosis rate. HCMs treated with the mimic showed a higher p53/pp53 ratio of 0.81 ± 0.17 than cells transfected with the scrambled siR, as the latter had a reduced ratio of 0.66 ± 0.08 (Figure 5C). Although the result was not statistically significant, with a *p*-value of 0.055, it showed a clear tendency towards a reduced apoptosis rate in the presence of increased miR-30d-5p.

**Figure 1 cells-12-02369-f001:**
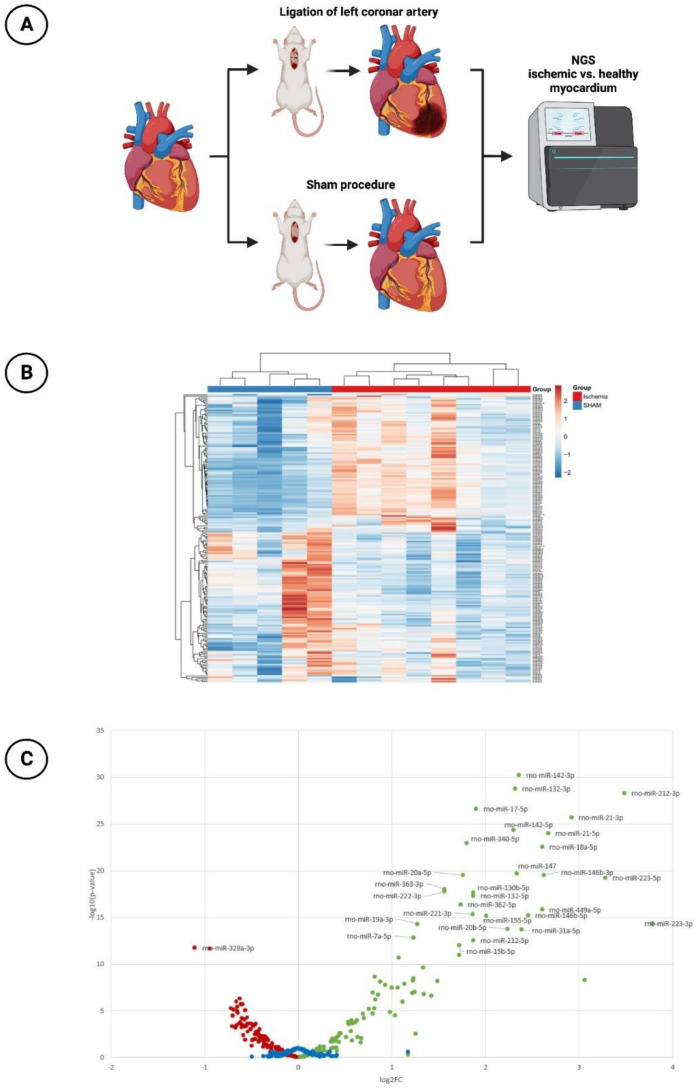
Next-generation sequencing (NGS) experiment. (**A**) Schematic overview of the rat model with subsequent NGS (*n* = 8 with LCA ligation; *n* = 5 with sham procedure). (**B**) Hierarchical clustering of investigated miRs (*n* = 202) in ischemic vs. healthy myocardium. (**C**) Volcano plot with corresponding log2(FC)−values (red dots: significantly downregulated miRs with negative log2(FC)−values; green dots: significantly upregulated miRs with positive log2(FC)−values; blue dots: miRs with non-significant changes in expression level).

**Figure 2 cells-12-02369-f002:**
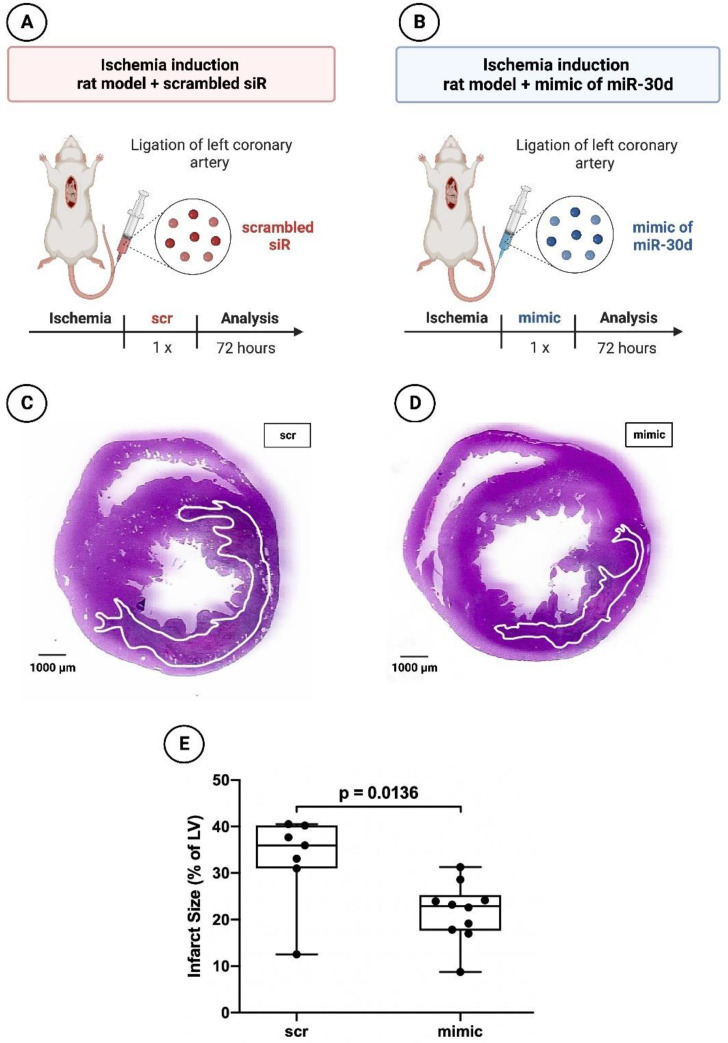
Planimetry of infarct size 72 h after induced myocardial infarction in experimental animal model using ligation of LCA. (**A**) Schematic overview of experimental rat model with administration of scrambled siR (*n* = 7). (**B**) Schematic overview of experimental rat model with administration of miR-30d-5p (*n* = 10). (**C**) Representative histological section (hematoxylin–eosin staining) through the myocardium of a 72 h animal with scrambled siR (scr) administration. (**D**) Representative histological section (hematoxylin–eosin staining) through the myocardium of a 72 h animal with miR-30d-5p (mimic) administration. (**E**) Boxplot diagram with corresponding *p*-value (Wilcoxon–Mann–Whitney U test) for a better comparison between intervention (mimic) and control groups (scr).

**Figure 3 cells-12-02369-f003:**
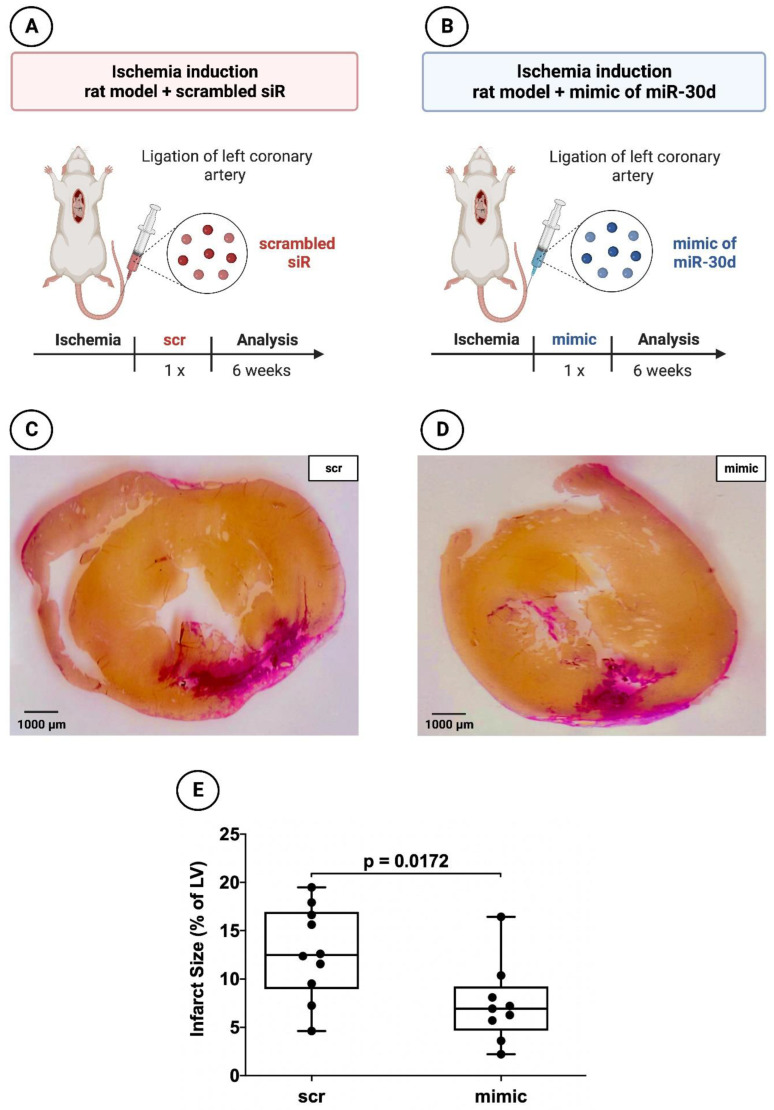
Planimetry of infarct size 6 weeks after induced myocardial infarction in experimental animal model using ligation of LCA. (**A**) Schematic overview of experimental rat model with administration of scrambled siR (*n* = 10). (**B**) Schematic overview of experimental rat model with administration of miR-30d-5p (*n* = 9). (**C**) Representative histological section (van Gieson staining) through the myocardium of a 6-week animal with scrambled siR (scr) administration. (**D**) Representative histological section (van Gieson staining) through the myocardium of a 6-week animal with miR-30d-5p (mimic) administration. (**E**) Boxplot diagram with corresponding *p*-value (Wilcoxon–Mann–Whitney U test) for a better comparison between intervention (mimic) and control groups (scr).

**Figure 4 cells-12-02369-f004:**
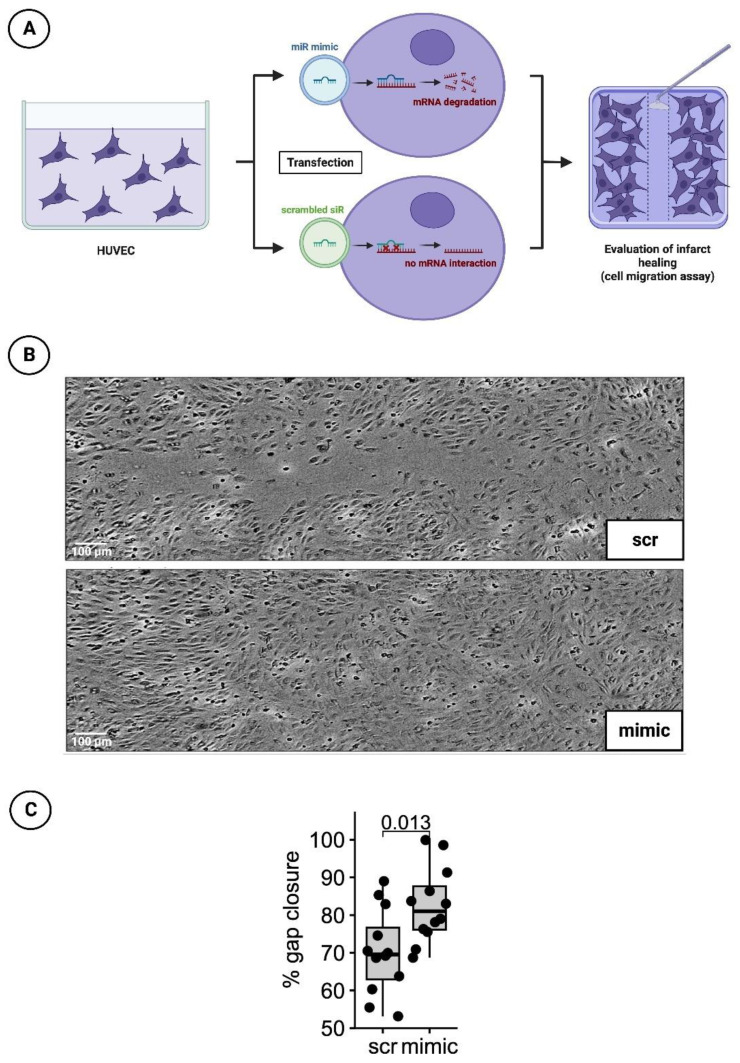
HUVEC cell migration scratch assay. (**A**) Schematic overview of cell migration scratch assay using HUVECs. (**B**) Representative histological images of scratch assays treated with scrambled siR (scr) or miR-30d-5p (mimic). (**C**) Boxplot diagram with corresponding *p*-value (Wilcoxon signed-rank test) for a better comparison between intervention (mimic) and control groups (scr).

**Figure 5 cells-12-02369-f005:**
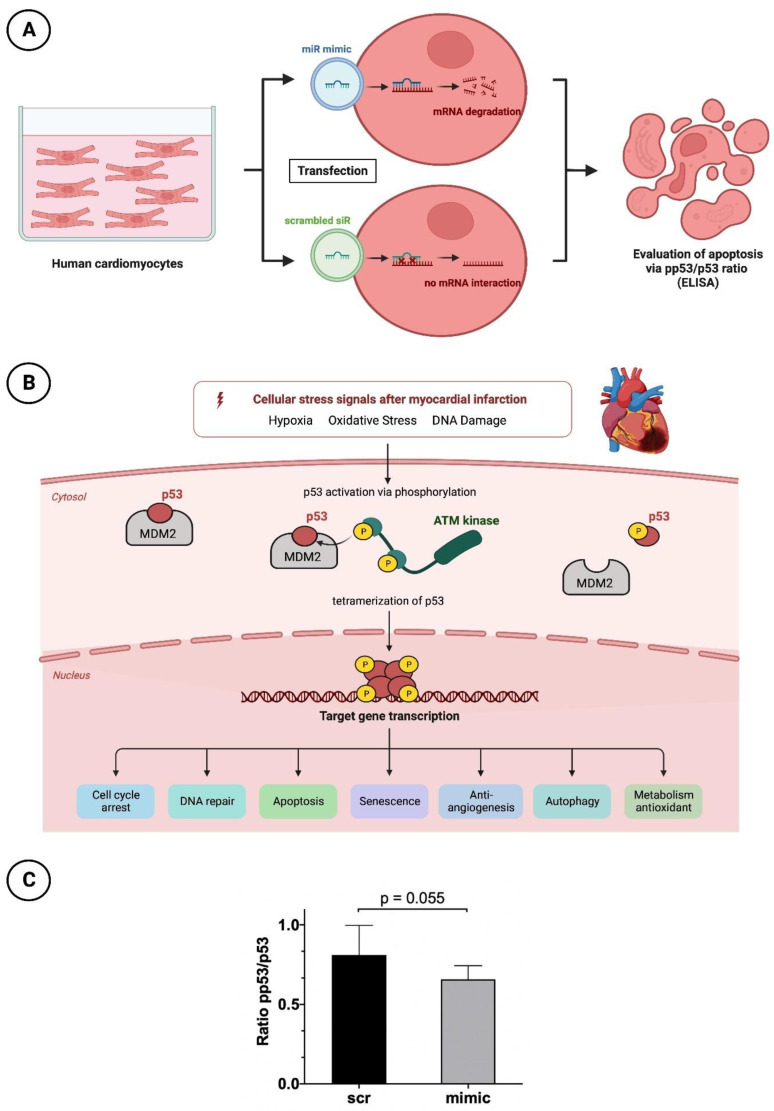
Apoptosis analysis in HCM with pp53/p53 ELISA. (**A**) Schematic overview of HCM treatment before apoptosis testing. (**B**) Very simplified representation of the apoptosis process in the context of myocardial infarction regarding p53 and pp53. (**C**) Bar chart with corresponding *p*-value (unpaired Student’s *t*-test) for a better comparison between intervention (mimic) and control group (scr).

## 4. Discussion

Myocardial ischemia in the context of MI leads to a significant restructuring of the miR expression profile. The alteration in the miR profile following ischemic MI represents a pivotal discovery with significant implications for advancing miR-based approaches to MI treatment. These changes reflect the heart’s response to ischemic injury, with specific miRs being upregulated or downregulated to modulate key pathways, such as inflammation, apoptosis, fibrosis, angiogenesis and hypertrophy [18]. This unique miR signature may hold potential as a diagnostic and prognostic marker for MI severity, offering non-invasive tools for assessment. Furthermore, understanding these miR alterations identifies promising therapeutic targets for miR-based interventions that can mitigate adverse cardiac remodeling and promote repair, ultimately improving cardiac function post-MI. Ongoing research in this area is critical for advancing our understanding of MI pathophysiology and the development of innovative miR-based therapies [40].

As shown in this work, cardiomyocytes stressed by hypoxia respond in particular with a marked upregulation of miRs measuring approximately 18–25 nucleotides in contrast to vital cardiomyocytes. The NGS performed here mainly detected miR-21, miR-146b and miR-223-3p as highly significant upregulated miRs associated with MI. MiR-21 is one of the most consistently upregulated miRNAs post-MI. It is involved in promoting cardiac fibrosis, hypertrophy and inflammation by targeting anti-fibrotic and anti-hypertrophic genes. Its upregulation is associated with adverse cardiac remodeling, which can worsen heart function over time [41]. MiR-146b is another upregulated miRNA in MI, primarily recognized for its role in modulating the inflammatory response. It targets genes are involved in inflammation pathways, potentially positively influencing the extent of inflammatory damage post-MI [42]. Elevated miR-146b levels may contribute to the regulation of pro-inflammatory signaling. MiR-223-3p is upregulated in MI and has been linked to the regulation of neutrophil function. It may play a role in modulating the inflammatory response by influencing the recruitment and activation of neutrophils in the infarcted myocardium [43].

A significantly lower number of miRs are downregulated in ischemic myocardial territory, but definitely not to the same extent as other miRs are upregulated (compare the corresponding logFC values of this study). Nevertheless, the focus in this experimental work was on the downregulated miRs to compensate for the resulting deficiency by therapeutic delivery. Based on the miR expression profile as well as the current literature, miR-30d-5p was chosen for further experimental study.

### 4.1. Anti-Remodeling Therapy—A Therapeutic Option to Reduce the Extent of Ischemia

After surviving a myocardial infarction with successful recanalization of the coronary vessel, the prevention of early and late complications plays a relevant role. The most frequent complication is the development of ischemic CMP with loss of myocardial contractility. Therefore, close echocardiographic follow-up is crucial to assess LVEF and to initiate drug therapy. In myocardial infarction patients, in addition to dual anti-platelet therapy and the use of statins, a drug combination therapy of beta-blockers and ACE inhibitors or AT1 antagonists is established irrespective of the LVEF in order to improve cardiac perfusion and simultaneously avert ventricular remodeling [44]. Ventricular remodeling is the term used to describe reactive remodeling processes of the non-infarcted ventricular myocardium as a late complication of myocardial infarction, which can manifest as fibrosis, hypertrophy and ventricular dilatation [45]. The exact pathomechanism is not clear at the present time; however, it is thought to be related to an exaggerated function of the renin–angiotensin–aldosterone system (RAAS). This at least explains the cardioprotective effect of the use of ACE inhibitors or AT1 antagonists. Also, miRs seem to play a significant role in post-myocardial infarction remodeling [46,47]. Maries et al. [48] showed in a review that numerous miRs (miR-1, -21, -29, -30, -34, -92, -133, -146a, etc.) were essential regulators of cardiac remodeling after myocardial infarction. miRs, however, not only offer therapeutic approaches with regard to successful anti-remodeling, but also have much more to offer at the molecular level.

### 4.2. What Do miRs Offer Regarding other Therapeutic Options in the Course of MI?

Since their discovery in the 1990s, miRs could open up a broad field of treatment for patients after myocardial infarction. An appealing compilation of the potential mechanisms of action or repair is provided by the reviews of Boon et al. [49] and Wang et al. [50]. These papers demonstrate that miRs have additional therapeutic docking sites for the prevention of ischemic CMP besides the aforementioned anti-remodeling. For example, significant anti-apoptotic potential and thus increased cardiomyocyte survival could be achieved by both regular, pathophysiological changes in miR expression profile and drug-induced up- or downregulation of specific miRs. Stimulation of cardiomyocyte proliferation and neovascularization of the infarct area by induced angiogenesis are also relevant mechanisms for the cardioprotective effect of different miRs. Finally, it should also be noted that a certain cardiac regenerative capacity could be achieved in in vitro studies by stimulation of pluripotent stem cells with certain miRs.

### 4.3. MiR-30d-5p and its Role in Ischemic CMP

MiR-30d-5p plays a role in underlying cardiovascular diseases that should not be underestimated. Xiao et al. [51] demonstrated in patients with acute heart failure that subjects with low serum levels of miR-30d-5p had a significantly higher 1-year mortality. Similarly, in a publication by Melman et al. [52], low circulating levels of miR-30d-5p were shown to be associated with a poorer response to cardiac resynchronization therapy.

In the context of acute MI and its sequelae, such as ischemic CMP, the current number of studies is scarce. Jia et al. [53] described a significant increase in the serum concentration of miR-30d-5p in the initial phase of MI (0–6 h) in infarcted patients in contrast to healthy subjects and even suggested a diagnostic superiority over the clinically used troponin I. After 4–6 h, the peak of the miR-30d-5p plasma level manifested itself with subsequently declining values. Bukauskas et al. [54] demonstrated in a direct comparison of serum levels of miR-30d-5p that ST segment-elevation myocardial infarction patients had significantly decreased levels 24 h after the event compared to healthy subjects. Regarding the expression of miR-30d-5p in the acute phase of MI, no conclusion could be drawn in our study because the animal model was deliberately designed for the development of ischemic CMP or ventricular remodeling.

However, it could be shown that after 6 weeks of “conservative therapy” of myocardial infarction in the animal model, a significant reduction in miR-30d-5p occurred in the myocardium. This was also demonstrated in an experimental work by Li et al. from 2021 [55], who associated decreased expression of miR-30d-5p with a significantly higher risk of adverse cardiac remodeling, fibrosis and inflammation. In another work by Li et al. from 2022 [56], the results could be applied to hypertrophic CMP.

From a pathophysiological standpoint, the initial release of miR-30d-5p into the systemic circulation stems from cardiac tissue in response to a lack of oxygen in the context of MI and the consequent tissue damage. The body’s reaction involves triggering inflammation and initiating healing processes, which help maintain elevated levels of miR-30d-5p as it plays a role in regulating inflammation and facilitating tissue repair. As the acute phase of MI subsides and the heart tissue stabilizes, there is a decrease in the release of miR-30d-5p, indicating a reduction in cellular stress and damage. The drop in circulating miR-30d-5p levels may also be associated with ongoing pathophysiological alterations in the heart, such as unfavorable restructuring, remodeling and fibrosis. These persistent changes within the cardiac environment can impact the production and release of miR-30d-5p in cardiomyocytes, potentially providing an explanation for the decreased availability of miR-30d-5p in patients with ischemic CMP both in the heart tissue itself and in the bloodstream.

### 4.4. Resistance to Ischemia, Anti-Apoptosis and Proliferation—Mechanisms of Action through Therapeutic Increase in miR-30d-5p

In the present study, serum concentrations were therapeutically increased by transfection of a mimic of miR-30d-5p in animal models (induction of ischemic CMP by ligation of LCA) and in different cell cultures (HCMs and HUVECs), and complex processes were investigated in vivo and in vitro. Finally, clinically relevant mechanisms of action could be deduced which might be causative for a possible prevention of ischemic CMP after MI.

In particular, in the animal model of induced ischemic CMP, a significant reduction in the infarct area could be demonstrated by increasing the serum concentration of miR-30d-5p, suggesting an increased stress resistance of cardiomyocytes regarding hypoxia. This statement confirmed the observations of Li et al. in 2021 [55], who also described a cardioprotective function against hypoxic stress in their animal models of ischemic CMP by increased accumulation of miR-30d-5p in cardiomyocytes.

A major mechanism behind this may lie in the regulation of autophagic and apoptotic processes, as miR-30d-5p has already demonstrated its potential in several previous ischemia studies in both the heart and brain. For example, Jiang et al. [57] demonstrated that in a rat model of induced acute cerebral ischemia, the administration of exosome-containing miR-30d-5p could significantly reduce the infarct area by suppressing autophagy. A reduction in autophagic processes by the administration of an agomir of miR-30d-5p was observed in developing rat brains after the induction of hypoxic–ischemic injury by Zhao and colleagues [58]. A similar constellation manifested in the heart, where in an MI-induced animal model, overexpression of miR-30d-5p decreased autophagy processes by reducing the activity of downstream autophagy-related 5 (ATG5) protein [59]. A definitive reduction in pro-apoptotic factors such as caspase 3 by miR-30d-5p was demonstrated in an in vitro work by Kim et al. [60] in ventricular cardiomyocytes. In the current work, the anti-apoptotic effect of miR-30d-5p was demonstrated once again, as HCMs treated with a mimic of miR-30d-5p already showed a significantly reduced apoptotic rate in the physiological state in contrast to HCMs transfected with scrambled siR.

Finally, the proliferation- and migration-inducing mechanism of miR-30d-5p should not be ignored. In contrast to numerous cancers, such as renal cell, pancreatic or colorectal carcinoma [61,62,63], in which an increase in miR-30d-5p causes suppression of the proliferation, migration and invasion of cancer cells, the situation in the heart is not entirely clear. Li et al. [55] described an inhibition of cardiac fibroblast activity by increased miR-30d-5p expression in their animal model and thus a reduction in cardiac fibrosis after induced MI. However, this contrasts with the findings of our HUVEC experiments with the cell migration scratch assays performed, in which cells with transfection of miR-30d-5p exhibited significantly faster gap closure and thus migration/proliferation. The extent to which miR-30d-5p acts in a cell-specific manner and is inductive for the migration of cardiomyocytes or endothelial cells into the newly formed infarct area in case of MI will have to be clarified in further in vitro and in vivo studies.

## 5. Limitation

Here, in this study we sought to analyze the pattern of miR alterations caused by myocardial ischemia. We furthermore sought to determine possible targets of therapeutic intervention and selected miR-30d-5p as a possible target. We are aware of the fact that myocardial ischemia affects miR signalling in a very broad fashion and that hundreds of miRs are affected by ischemia. Our intention was to focus on the effects of one single miR in this pathophysiological condition and to determine possible mechanistic effects in in vitro experiments. We want to propose miR-30d-5p as an interesting target molecule in myocardial ischemia, though we know that this study can just give hints for possible mechanisms and that further in vitro and in vivo studies are necessary to gain more insight into cellular mechanisms in order to gain more knowledge for possible therapeutic use.

## 6. Conclusions and Outlook

Using a mimic of miR-30d-5p underlines the cardioprotective effects of miR-30d-5p in MI/IR and could reduce the risk for ischemic CMP development. Future studies should investigate the clinical efficacy and safety of miR-30d-5p mimic therapy in human MI patients, assessing its long-term impact on cardiac outcomes. Additionally, mechanistic research should focus on elucidating the molecular pathways and downstream targets of miR-30d-5p to better understand its cardioprotective mechanisms. And finally, exploring potential synergies with established MI treatments and optimizing delivery methods for miR-30d-5p mimic therapy may enhance its therapeutic potential in cardiac remodeling.

## Figures and Tables

**Table 1 cells-12-02369-t001:** Tabular overview of 20 best upregulated miRs based on log2(FC)-values.

miRNA	log2(FC)	*p*-Value
rno-miRNA-223-3p	3.78	4.64 × 10^−15^
rno-miRNA-212-3p	3.48	4.49 × 10^−29^
rno-miRNA-223-5p	3.28	5.71 × 10^−20^
rno-miRNA-3473	3.06	5.27 × 10^−09^
rno-miRNA-21-3p	2.92	1.96 × 10^−26^
rno-miRNA-21-5p	2.67	9.28 × 10^−25^
rno-miRNA-146b-3p	2.62	2.99 × 10^−20^
rno-miRNA-449a-5p	2.61	1.47 × 10^−16^
rno-miRNA-18a-5p	2.61	2.71 × 10^−23^
rno-miRNA-146b-5p	2.46	5.72 × 10^−16^
rno-miRNA-31a-5p	2.39	2.07 × 10^−14^
rno-miRNA-142-3p	2.36	6.02 × 10^−31^
rno-miRNA-147	2.34	2.09 × 10^−20^
rno-miRNA-132-3p	2.32	1.69 × 10^−29^
rno-miRNA-142-5p	2.30	4.14 × 10^−25^
rno-miRNA-20b-5p	2.24	1.82 × 10^−14^
rno-miRNA-155-5p	2.01	7.11 × 10^−16^
rno-miRNA-17-5p	1.90	2.47 × 10^−27^
rno-miRNA-212-5p	1.87	2.75 × 10^−13^
rno-miRNA-221-3p	1.86	4.37 × 10^−16^

**Table 2 cells-12-02369-t002:** Tabular overview of 20 best downregulated miRs based on log2(FC)-values.

miRNA	log2(FC)	*p*-Value
rno-miRNA-328a-3p	−1.10	1.79 × 10^−12^
rno-miR-125a-5p	−0.94	2.04 × 10^−12^
rno-miR-874-3p	−0.70	4.85 × 10^−04^
rno-miRNA-133-5p	−0.69	3.46 × 10^−05^
rno-miRNA-100-5p	−0.68	2.92 × 10^−08^
rno-miR-325-5p	−0.67	6.82 × 10^−04^
rno-miRNA-351-5p	−0.66	1.14 × 10^−06^
rno-miRNA-30d-5p	−0.65	4.27 × 10^−06^
rno-miRNA-128-3p	−0.65	1.29 × 10^−05^
rno-miRNA-1-3p	−0.64	1.72 × 10^−05^
rno-miR-150-3p	−0.63	2.49 × 10^−04^
rno-miRNA-26a-5p	−0.62	5.16 × 10^−07^
rno- miR- 30e- 5p	−0.62	1.64 × 10^−04^
rno-miR-125b-5p	−0.61	2.00 × 10^−06^
rno-miRNA-22-3p	−0.60	3.12 × 10^−04^
rno-miRNA-181d-5p	−0.59	2.24 × 10^−06^
rno-miR-208b-3p	−0.59	4.62 × 10^−03^
rno-miR-22-5p	−0.58	5.24 × 10^−03^
rno-miR-145-3p	−0.56	9.70 × 10^−06^
rno-miRNA-664a-3p	−0.55	4.76 × 10^−04^

## Data Availability

The data presented in this study are available on request from the corresponding author.

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
