# Peer review of "MicroRNA-30d-5p—A Potential New Therapeutic Target for Prevention of Ischemic Cardiomyopathy after Myocardial Infarction"

_cells, 2023, doi:10.3390/cells12192369_

Round 1
Reviewer 1 Report
Dear Authors,
It is very interesting and important paper to develop microRNA(miR) therapy for ischemic cardiomyopathy after myocardial infarction. Study was done by using animal model and in vitro culture system. For making more attractive paper, revising of structure itself of the manuscript, especially clearer description of the methodology, are recommended.
A. Major query and comments:
1. Coordination between findings of array analysis and findings of miR-30d experiments should be presented and discussed more clearly.
1.1 Please explain the reasons to select miR-30d as a candidate miR of biomarker and therapeutic agent.
1.2 Please give more discussion about upregulated miRs
1.3 Make sure that the authors used miR-30d-5p or miR-30d-3p: both were downregulated in the NGS analysis. However, no obvious description whether miR-30d-5p or miR-30d-3p was used in the following experiments including “mimic”. It is better to show the sequence or GENE ID (Line 112 Fig 1E). Only in Lines 299 to 307, they used “miR-30d-5p”. The author should be consistent to use annotation of miRNA.
2. Material and Methods:
The structure of this manuscript is not well regulated. Material and Methods contain many important and interesting results.
2.1 Line 287: Fig 1B, C, D should be moved to in “Results”.
2.2 Line 393-415: It is not easy to understand the linkage among down-regulation of miR-30d in heart tissue, low circulating levels of miR-30d, mortality caused by MI. Clearer explanation is better for readers. Lower levels of miR-30d in ischemic heart tissue and mimic inhibit tissue damage in vivo showed good correlation. How does the author correlate these finding with previous observation that circulating miR-30d was downregulated in MI patients. What is the origin of circulating miR-30d in normal and MI patients.
3. Interpretation of findings
One MicroRNA does not act on the target gene like as one gene-one mRNA-one protein relationship, called as “Central Dogma”. MiRNA acts to hundreds target genes and one gene function under control of multiple miRNAs, namely miRNA system is acting under the network out of “Central Dogma”. In this point of view, this study was done by using the results from array analysis. It is very precious work.
3.1 Please give some comments on the change of the profile itself of miRNAs in heart after ischemic insult. That would be an important finding for further development of miRNA approach to MI.
3.2 Lines 73 to 75: The authors show previous studies reporting the multiple effective miRNAs (-21, -199a, -590, -1, -24, -132) in cardiology. Findings about these important miRNAs obtained from the array analysis in this study should be shortly discussed.
3.3 Is there any evidence to show transfer of injected miR-30d into cardiomyocytes after intraperitoneal injection in this study or previous reports?
3.4 Question about specificity of transportation of miR-30d. The author should show cardiomyocyte-specific transfer of miR-30d, if possible.
3.5 Please give discussion on what was the cause of down-regulation of miR-30d, or which cells is the origin of circulating miR-30d. This may give us more understanding of pathophysiological role of miR-30d in MI.
4. Comments on delivering system for miR theray should be given.
B. Minor comments:
1. Line 112: Figure 1 contain experimental design and NGS results, at the same time. These should be separated. Results should be described in the section of “Results”.
2. Line 250: anti-apopteutic to “apoptotic”.
3. Lines 323-325: Staining methods of section in Fig. 3 looks totally different from section in Fig. 2. Please explain what difference of staining methods between Fig. 2 and 3 is.
4. Line 432: “exosomes of miR-30d” should be “exosome containing miR-30d”
5. Line 239: “controlled cell death” may be “programmed cell death”
6. Lines 292-296: More clear explanation on drug-induced up-regulation is needed. Why the author decided these up-regulation is the effect of medication.
(revised version by reviewer 20230921)
Minor revision is enough. Please refer to reviewer's comments.
Author Response
Dear Reviewer 1,
thank you very much for the valuable comments on our manuscript. Below you will find a PDF file in which we describe in detail our manuscript changes based on your suggestions.
Kind regards
Dr. Elke Boxhammer

Reviewer 2 Report
In this study, Boxhammer et.al investigated the therapeutic potential of miR-30d in improving cardiac remodeling after myocardial infarction (MI)
Overall, this study has an interesting topic, good rationale and clear hypothesis, detailed method section and appropriate references. However, it can be further improved by addressing the comments below:
1. Method section contains very detailed information which can be moved to result section. Figure legend should contain more details, for instance, #animals per group, what staining used, what is the experimental condition, statistical analysis etc.
2. It is still kind of confusing why the selection of miR-30d for further investigation was based on down-regulation alone, what is the rationale for choosing miR-30d over other more significant down-regulated miRNAs.
3. Figure 1, the quality of figures should be improved, for instance, nothing is readable in figure 1b.
4. In figure 1c, different color could be used to distinguish significantly upregulated, down regulated genes and gens are not significantly between two groups. Representant down regulated genes should also be annotated in the plot. More detailed figure legend is needed.
5. Figure 2C-D, it is not clear where is the infarct area, better representative images should be used. What staining is used for figure 2C-2D and 3C-3D, and how many animals per group? detailed information should be included in figure legend.
6. Figure 2 and figure 3 should include each individual data point in the boxplot.
7. It is very nice to include a limitation section, it would be helpful to include more specific suggestions for future directions in this section. Like which aspects should be explored in subsequent in vivo or clinical studies.
8. To evaluate apoptosis, p53/pp53 ELISA was performed, other apoptosis method should be further explored like Caspase Activity Assays, TUNEL Assay or MTT assay
Moderate editing of English language required
Author Response
Dear Reviewer 2,
thank you very much for the valuable comments on our manuscript. Below you will find a PDF file in which we describe in detail our manuscript changes based on your suggestions.
Kind regards
Dr. Elke Boxhammer

Reviewer 3 Report
This is a well written study showing that using a mimic of miR-30d enhanced cardioprotective effects of miR-30d in a rat model of MI and MR. In vivo results were supported by using cell culture assays (gap closing, apoptosis rate).
The presented results offers a therapeutic potential even in men. Mircorna treatments are a promising approach in future clinical studies.
Minor concern: Please include the dose finding of the rna´s used.
Author Response
Dear Reviewer 3,
thank you very much for the valuable comments on our manuscript. Below you will find a PDF file in which we describe in detail our manuscript changes based on your suggestions.
Kind regards
Dr. Elke Boxhammer

Round 2
Reviewer 2 Report
The authors has addressed all of my comments.
Minor editing of English language required